# Understanding the contribution of public- and restricted-access places to overall and domain-specific physical activity among Mexican adults: A cross-sectional study

**Alejandra Jáuregui**[1]*, **Deborah Salvo**[1,2], **Catalina Medina**[1], **Simón Barquera**[1], **David Hammond**[3]

**1** Center for Nutrition and Health Research, National Institute of Public Health, Cuernavaca, Morelos, Mexico, **2** Washington University in St. Louis, Brown School, Prevention Research Center & Center for Diabetes Translation Research, Saint Louis, Missouri, United States of America, **3** School of Public Health and Health Systems, University of Waterloo, Waterloo, Ontario, Canada

* alejandra.jauregui@insp.mx

**Data Availability Statement:** The database used for this analysis is available from https://doi.org/10.6084/m9.figshare.11803488.v1.

## Abstract

Latin Americans engage in physical activity (PA) in unique ways and use a wider range of places for PA than those commonly studied in high-income settings. We examined the contribution of a variety of places and domains of PA to meeting PA recommendations among a sample of adults (18–65 y) from all over Mexico. This was a cross-sectional study conducted in 2017 (n = 3 686). Overall and domain-specific PA was measured using the Global Physical Activity Questionnaire. Use of places for PA was self-reported. Places were classified as private or public. In 2018, associations between specific places and meeting PA recommendations (≥150 mins/week) were estimated using multivariate logistic regression models. In total 72.1% met PA recommendations. The proportion meeting recommendations through domain-specific PA was highest for leisure-time PA (50.0%), followed by travel-related (39.1%) and work-related (24.9%) PA. The most commonly reported places for PA were home (43%), parks (40.7%) and streets (39.4%) (public). Use of most public places was positively associated with meeting PA recommendations, mainly through travel-related PA (Streets OR 2.05 [95% CI 1.71–2.45]; Cycling paths OR 1.91 [1.37–2.68]). Using private places was more strongly associated with PA, mainly leisure-time PA (Gyms OR 9.66 [7.34–12.70]); Sports facilities OR 5.03 [3.27–7.74]). In conclusion, public and private places were important contributors to PA. While public places may be a powerful setting for PA promotion, increasing the equitable access for all to private places may also represent an effective strategy to increase PA among Mexican adults.

## Introduction

Latin America has the highest prevalence of physical inactivity among adults relative to all other world regions [1]. In Mexico, every 1 in 4 men and 1 in 3 women are physically inactive,

**Funding:** Funding for this project was provided by a Canadian Institutes of Health Research (CIHR) Project Grant, with additional support from an International Health Grant, the Public Health Agency of Canada (PHAC), and a CIHR – PHAC Applied Public Health Chair (Hammond).

**Competing interests:** The authors have declared that no competing interests exist.

and the prevalence of physical inactivity has increased in the past 10 years [2]. Given the magnitude of this public health crisis, international organizations have made an urgent call to prioritize policies and actions for promoting physical activity (PA) [3]. Increasing the availability, equitable access, and quality of places where people are likely to be physically active is a priority investment that works for PA promotion [3]. However, a major limitation for implementing effective PA promotion policies at scale is the lack of contextually-relevant evidence on the underlying drivers of physical (in)activity among Mexicans, as well as other Latin American countries [4,5].

Latin Americans engage in PA in unique ways. Studies suggest that in comparison to high-income countries, Latin Americans engage in more transport-based PA, and less leisure-time PA, and use a wider range of places for PA (e.g., non-exercise or sport oriented public-access places like shopping malls, public squares, or unoccupied parcels/lots) [6]. A study that measured walking behaviors among adults from 12 countries, including three Latin-American sites (Mexico, Colombia and Brazil), found that two of the three Latin-American sites ranked amongst the highest for engagement in any walking for transport, and the three of them were the lowest ranked for the prevalence of any walking for leisure [4]. The widest disparities were observed in Cuernavaca, a middle-income Mexican city where >90% reported transport-walking, and only 35% reported leisure-walking [1,4]. This study also reported that almost half of adults from Cuernavaca, achieved ≥150 weekly minutes of transport-based PA, whereas only 25% did so for leisure-time PA [7]. Similar patterns have been reported in other Latin American urban settings including Bogota, Colombia, and Curitiba, Brazil [4].

A study of three Latin-American cities exploring the use of public and private places for PA found that those who frequently used public places were more likely to be physically active, mainly by walking [6]. However, this study was limited in that it only examined composite variables of public versus private places (with no analysis of specific places). It also only examined the relation of use of places with overall- and leisure-time PA, without considering other domains of PA such as transport- or work-related PA. Another limitation was that it only included one city from each of the studied countries (i.e. Curitiba, Brazil, Bogotá, Colombia, and Cuernavaca, Mexico), which does not represent the full spectrum of urban settings in a country.

We examined the contribution of a variety of places and domains of PA to meeting PA recommendations among a sample of adults from all over Mexico. Understanding the contributions of specific places and domains of PA to meeting PA recommendations is crucial to implement effective PA promotion strategies in Mexico. Insights into potential places for PA promotion in a country with a large geopolitical region with highly diverse physical, social, cultural, and economic characteristics such as Mexico could guide future strategies in similar Latin-American regions.

## Materials and methods

### Study design

This study is part of the International Food Policy Study, a study conducted in five countries, including Mexico. The current analysis was conducted in 2018 and used cross-sectional data collected in December 2017. This study was approved by the Research Ethics Committee of the University of Waterloo (ORE # 21460).

### Participants and data collection

Data were collected via self-completed web-based surveys with adults aged 18–64 years. Respondents were recruited through the Nielsen Consumer Insights Global Panel and partner

panels. Individuals were eligible to participate if they were ≥18 years of age. Participants were recruited using both, probability and non-probability sampling methods [8]. For the current project, Nielsen targeted recruitment, so that the percentage of participants in each age group would be similar to the general population in Mexico. Email invitations with unique survey access links were sent to a random sample of panelists within the specified age. Among 68,336 email invitations sent, 4,057 surveys were completed (response rate 5.9%). A total of 33% of participants lived in Mexico City or Mexico State, whereas the rest lived in cities located in the 30 other states in Mexico. Respondents provided written consent prior to completing the survey. Participants were more highly educated and had a lower self-reported BMI compared to national estimates [9,10].

## Physical activity measures

PA was measured using the Global Physical Activity Questionnaire (GPAQ). GPAQ is comprised of 19 questions measuring work-related, transport-related and leisure-time PA. This instrument has moderate to fair validity and good reliability across international and Latin American populations [11–14]. Data were processed in accordance to the GPAQ Analysis guide [15]. After eliminating implausible (e.g. reporting more than 16 hours per day for a specific sub-domain of PA) and inconsistent (e.g. reporting zero days for a specific sub-domain but but values >0 in the corresponding time variable) PA data, MET-minutes per week for overall and domain-specific PA were calculated. Participants were classified as meeting PA recommendations if they had 600 MET-min-week or more (equivalent to 150 minutes per week of MVPA). Additionally, we constructed domain-specific binary outcome variables for meeting PA recommendations through specific PA domains. We assessed this as achieving 600 MET-min-week or more of travel-related, work-related, and leisure-time PA.

## Places for physical activity

The use of a set of 15 formal and informal places for PA was ascertained by questionnaire. These questions were developed and tested by expert Latin-American PA researchers from three countries and used in the International Physical Activity and Environment Network study in Latin American sites [4,6]. Formal places included parks, cycling paths, private courts or sports facilities, private gyms, school or university campuses, outdoor green spaces, and public recreation centers. Informal places included plazas (public squares), informal outdoor courts (undeveloped land), streets (includes sidewalks), home, shopping malls, bars and nightclubs, and museums. Participants were asked to indicate the three most common places that they use to be physically active.

Places were classified as public (no fee or membership required: parks, streets, outdoors/open green spaces, shopping malls, outdoor courts, plazas, cycling paths, and indoor courts) or private places (requiring a fee or membership: private gyms, private sports facilities, school/university, bars and nightclubs and museums) [6]. 'Home' and work were not included in the "private places for PA" category, since gaining access to one's own home or work generally does not require any special access permit. Participants were classified as using any public place (yes/no) or any private place (yes/no) for PA.

## Covariates

Self-reported sociodemographic information (e.g. gender, age, education level, ethnicity, working status, children living in the household—see Table 1). Income adequacy was assessed with the question '*Thinking about your total monthly income, how difficult or easy*

**Table 1. Demographic characteristics of the sample (n = 3,686).** Mexico, 2017.

|  | % (95% CI) [a] |
|---|---|
| **Gender** | |
| Female | 51.9 (50.0, 53.8) |
| Male | 48.1 (46.2, 50.0) |
| **Age group** | |
| 18–24 y | 19.8 (18.5, 21.1) |
| 25–30 y | 16.6 (15.4, 17.8) |
| 31–39 y | 21.1 (19.7, 22.5) |
| 40–49 y | 21.2 (19.7, 22.9) |
| 50–59 y | 16.8 (15.1, 18.7) |
| 60–64 y | 4.5 (3.6, 5.6) |
| **Education level** | |
| High school or less | 16.5 (15.2, 17.9) |
| Undergraduate level | 68.7 (66.9, 70.4) |
| Graduate level | 14.7 (13.4, 16.1) |
| **Ethnicity** | |
| Majority group | 87.4 (86.1, 88.6) |
| Indigenous | 12.6 (11.4, 13.9) |
| **Children living in the household** | |
| Yes | 50.7 (48.8, 52.5) |
| No | 49.3 (47.5, 51.2) |
| **Body mass index category** | |
| Underweight | 2.5 (2.0, 3.1) |
| Desirable weight | 42.7 (40.8, 44.5) |
| Overweight | 35.1 (33.3, 36.9) |
| Obesity class I | 12.7 (11.5, 14.0) |
| Obesity class II or more | 7.0 (6.1, 8.1) |
| **Income adequacy** | |
| Difficult | 41.7 (39.8, 43.6) |
| Neither easy nor difficult | 38.2 (36.4, 40.0) |
| Easy | 20.1 (18.7, 21.6) |
| **Region of the country** | |
| North | 23.2 (21.8, 24.8) |
| South | 29.4 (27.6, 31.3) |
| Center | 39.2 (37.4, 41.1) |
| Mexico City | 8.2 (7.6, 8.8) |

a. Estimations (% and 95% CI) are weighted using post-stratification survey weights

*is it for you to make ends meet*?', with responses collapsed into difficult, neither easy nor difficult, or easy. This question has been used to evaluate the actual financial status in adults and household financial distress [16]. Body-mass-index (BMI) was estimated ($kg/m^2$) through self-reported weight and height and participants were classified as underweight ($18.5 \ kg/m^2$), normal weight (18.5 to 24.9 $kg/m^2$), overweight (25.0 to 29.9 $kg/m^2$), or obese ($> 30 \ kg/m^2$), according to the World Health Organization categories [17]. Regions of Mexico (North, South, Center and, Mexico City) were constructed according to the reported State of residency in Mexico. These regions are used in the country to capture cultural and economic differences in Mexico [10].

## Statistical analysis

All analyses were adjusted for post-stratification sampling weights based on population distributions of sex, age and country-region. Descriptive statistics (means, frequencies, and 95% CIs) were calculated for all variables. To identify the places where participants were physically active, we estimated the prevalence (95% CI) of reported use of each place for PA. Places were ranked according to their prevalence of use. PA and places were described in the entire sample and by demographic characteristics.

We used two sets of multivariate logistic regression models to examine the associations between specific places and PA outcomes (meeting PA recommendations through overall, travel-related, work-related, or leisure-time PA). The first set of models examined the association between PA outcomes and using any public place or any private place for PA. Models were adjusted for potential confounders, including sociodemographic characteristics (i.e. gender, age, education, ethnicity, children living in the household, and self-reported income adequacy), BMI, region of the country, and the use of home and work for PA. The second set of models examined associations between places and PA outcomes by introducing all individual studied places in the model. These models were adjusted for the same sociodemographic characteristics, BMI and region of the country. All models for travel-related, work-related and leisure time PA were additionally adjusted for meeting PA recommendations through the other two PA domains (e.g. the model for travel-related PA was adjusted for work-related and leisure-time PA). A p-value of 0.01 was considered statistically significant to account for the use of several models. All analyses were conducted using STATA version 14.0 (StataCorp Inc).

## Results

Of the 4,057 participants that completed the survey, 34 had implausible values or inconsistent answers for PA, and 337 had missing data for covariates (BMI = 206; ethnicity = 55; education level = 19; income adequacy = 33; student status = 5; working status = 1; having any child = 17), leaving 3,686 participants for analysis. No differences in demographic information were observed between excluded- and included participants. The weighted sample was fairly balanced with respect to gender, age, region of the country, and whether participants lived with children or not (Table 1). Most participants did not consider it difficult to make ends meet (58.3%), were employed full-time (60.1%), were university educated (83.4%), and were overweight or obese (54.6%).

The prevalence of meeting PA recommendations was of 72.1% (Table 2). Half of participants met PA recommendations through leisure-time, and 39.1 and 24.9% achieved recommendations through transport- and work-related PA, respectively. A higher proportion of participants with a lower education level met PA recommendations through travel-related and work-related PA (p<0.01), whereas more (60.8%) participants with a higher income adequacy met PA recommendations through leisure-time PA (p<0.01).

Home was the most frequently reported place for PA (43.4%), followed by four public places: parks (40.7%), streets (39.4%), other open green spaces (25.0%), and shopping malls (21.3%) (Fig 1). Overall, more participants reported using at least one public place for PA (78.6%) than at least one private place (35.4%) (Fig 1). Women's use of places for PA was generally lower than that of men for most places (Fig 2A). Participants with a higher educational attainment (Fig 2B) or a higher income adequacy (Fig 2C) reported a higher use of private places for PA, such as private gyms or sports facilities (p<0.01). The ranking of places and proportion of use across demographic characteristics are presented in S1 Table, S2 Table, and S3 Table.

**Table 2. Proportion of participants achieving physical activity recommendations among Mexican adults, by demographic characteristics (n = 3,686).**

|  | Overall PA | Travel-related PA | Leisure-time PA | Work-related PA |
|---|---|---|---|---|
|  | % (95% CI) [a] | % (95% CI) [a] | % (95% CI) [a] | % (95% CI) [a] |
| **Full sample** | 72.1 (70.3, 73.8) | 39.1 (37.3, 40.9) | 50.0 (48.1, 51.8) | 24.9 (23.3, 26.6) |
| **Gender** |  |  |  |  |
| Males | 74.6 (72.3, 76.8) | 42.6 (40.1, 45.1) | 55.2 (52.7, 57.7) | 26.8 (24.7, 29.1) |
| Females | 69.7 (67.1, 72.2)[b] | 35.9 (33.3, 38.5)[b] | 45.1 (42.4, 47.8)[b] | 23.1 (20.9, 25.5)[b] |
| **Age group** |  |  |  |  |
| 18–24 y | 75.3 (72.2, 78.2) | 40.6 (37.3, 44.1) | 52.5 (49.0, 55.9) | 31.4 (28.3, 34.7) |
| 25–30 y | 76.0 (72.8, 79.0) | 45.2 (41.7, 48.9) | 54.9 (51.3, 58.5) | 27.3 (24.2, 30.6) |
| 31–39 y | 70.4 (67.0, 73.7) | 36.4 (32.9, 40.0) [d] | 50.2 (46.6, 53.9) | 23.2 (20.3, 26.4) [c] |
| 40–49 y | 68.9 (64.7, 72.7) [d] | 37.1 (33.0, 41.3) [d] | 49.9 (45.6, 54.2) | 22.8 (19.4, 26.5) [c] |
| 50–59 y | 70.9 (65.0, 76.2) | 35.8 (30.3, 41.8) [d] | 41.9 (36.0, 47.9) [c,d] | 22.7 (18.1, 28.2) [c] |
| 60–64 y | 70.5 (59.1, 79.7) | 43.8 (33.2, 55.0) | 49.7 (38.6, 60.8) | 13.9 (7.8, 23.5) [c] |
| **Education level** |  |  |  |  |
| High school or less | 74.3 (70.2, 78.0) | 44.5 (40.2, 48.9) | 47.4 (43.0, 51.7) | 34.1 (30.1, 38.3) |
| Undergraduate | 72.1 (69.9, 74.1) | 39.2 (37.0, 41.5) | 50.2 (47.9, 52.5) | 24.6 (22.7, 26.5) [e] |
| Graduate | 69.7 (64.7, 74.2) | 32.3 (27.8, 37.3) [e] | 51.9 (46.7, 57.0) | 16.3 (12.9, 20.5) [e,f] |
| **Income adequacy** |  |  |  |  |
| Difficult | 68.1 (65.2, 70.8) | 37.5 (34.6, 40.4) | 42.6 (39.7, 45.6) | 25.2 (22.7, 27.7) |
| Neither easy nor difficult | 74.4 (71.7, 77.0) [g] | 41.4 (38.5, 44.4) | 52.3 (49.2, 55.3) [g] | 24.9 (22.3, 27.6) |
| Easy | 75.8 (72.0, 79.2) [g] | 38.0 (34.2, 42.0) | 60.8 (56.7, 64.8) [g,h] | 24.5 (21.2, 28.1) |

PA = Physical activity

a. Estimations (% and 95% CI) are weighted using post-stratification survey weights

b. Significantly different from males

c. Significantly different from 18-24y

d. Significantly different from 25-30y

e. Significantly different from high school or less

f. Significantly different from undergraduate level

g. Significantly different from difficult income adequacy

h. Significantly different from neither easy nor difficult income adequacy

Table 3 presents the adjusted associations between places for PA and achieving PA recommendations. The use of any public place or any private place was associated with 1.85 (1.49–2.30) and 3.29 (2.67–4.06) higher odds of meeting overall PA recommendations, respectively. The strongest associations for this outcome were observed for specific private places (i.e., Private gyms OR = 6.26 [4.52–8.68], Private sports facilities OR = 3.13 [1.85–5.28]) and public places (i.e. Outdoor courts OR = 2.97 [2.11–4.20], Parks OR = 2.31 [1.88–2.83], Cycling Paths OR = 2.03 [1.26–3.28]). In contrast, reporting use of shopping malls was associated with 43% (28–55%) lower odds of meeting overall PA recommendations.

The use of any public place for PA was associated with 2.41 (1.94–2.99) higher odds of meeting PA recommendations through travel-related PA. Specific places accounting for this association were streets (OR = 2.05 [1.71–2.45]), cycling paths (OR = 1.91 [1.37–2.68]), and parks (OR = 1.34 [1.12–1.59]. Meanwhile, the use of shopping malls was associated with 29% (11–43%) lower odds for this outcome.

Meeting PA recommendations through leisure-time PA was positively associated with the use of any private place (OR = 4.36 [3.62–5.25]). Specifically, private gyms (OR = 9.66 [7.34–12.70]) and private sports facilities (OR = 5.03 [3.27–7.74]) were strongly and positively

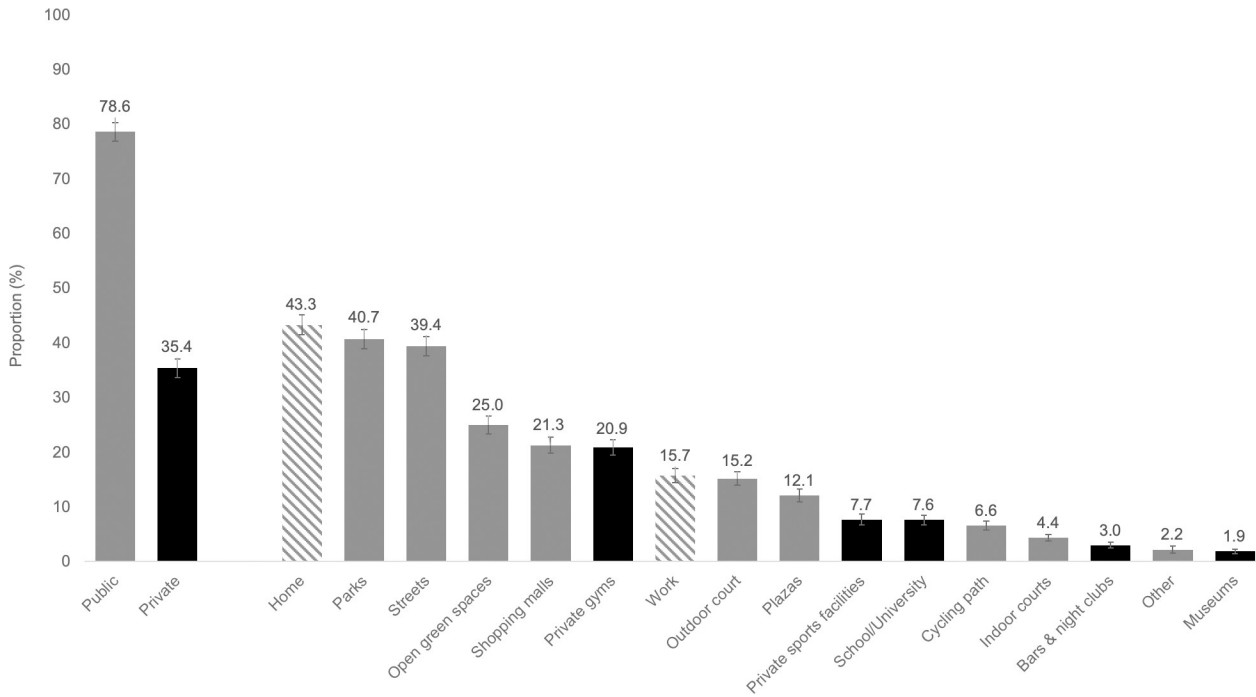

**Fig 1. Most frequently reported places for physical activity among adults from Mexico (n = 3,686).** Black: Open-access place (no cost, membership or affiliation may be required for access and use). Gray: Restricted-access place (cost, membership or affiliation required for access and use). Dashed bars: Home and work. Estimations (% and 95% CI) are weighted using survey weights.

associated to this outcome. Additionally, using one's home (OR = 1.45 [1.20–1.75]) and some public places were also positively (i.e., Outdoor courts OR = 3.48 [2.61, 4.64], Parks OR = 1.91 [1.57–2.32]) associated to meeting PA through leisure-time PA. Meanwhile, using work (OR = 0.45 [0.35–0.59]) and some specific public places (Streets OR = 0.73 [0.60–0.89], Shopping Malls OR = 0.58 [0.45–0.75]) were negatively associated with this outcome.

Neither using any public place nor using any private place were associated with meeting PA recommendations through work-related PA; using one's work (OR = 3.18 [2.49–4.06]) was the only specific place associated with this outcome.

## Discussion

We examined the contribution of specific places and domains of PA to meeting PA recommendations among a sample of adults from all over Mexico. We found that among our sample, most PA is due to either leisure-based or travel-related activity. Use of public places, such as parks or streets, was very high among all sub-groups of our sample, and were important contributors to meeting PA recommendations. Private places were used by less people, however, the use of these places was more strongly (higher estimate magnitudes observed) related to the studied PA outcomes, and in particular, to leisure-time PA.

Some of the observed PA patterns were consistent with findings from other studies of Latin American cities, including Bogota (Colombia), Curitiba (Brazil), and Cuernavaca (Mexico) [4,7]. Of particular relevance is that the contribution of active travel among this sample was considerably high, as reported for such cities [4]. Observed differences in the contribution of domain-specific PA by education and income adequacy level, two proxies of socioeconomic status, suggest that utilitarian PA (i.e. travel- and work-related PA) is an important contributor to overall PA, mainly among those of a lower economic position. In contrast, recreational and

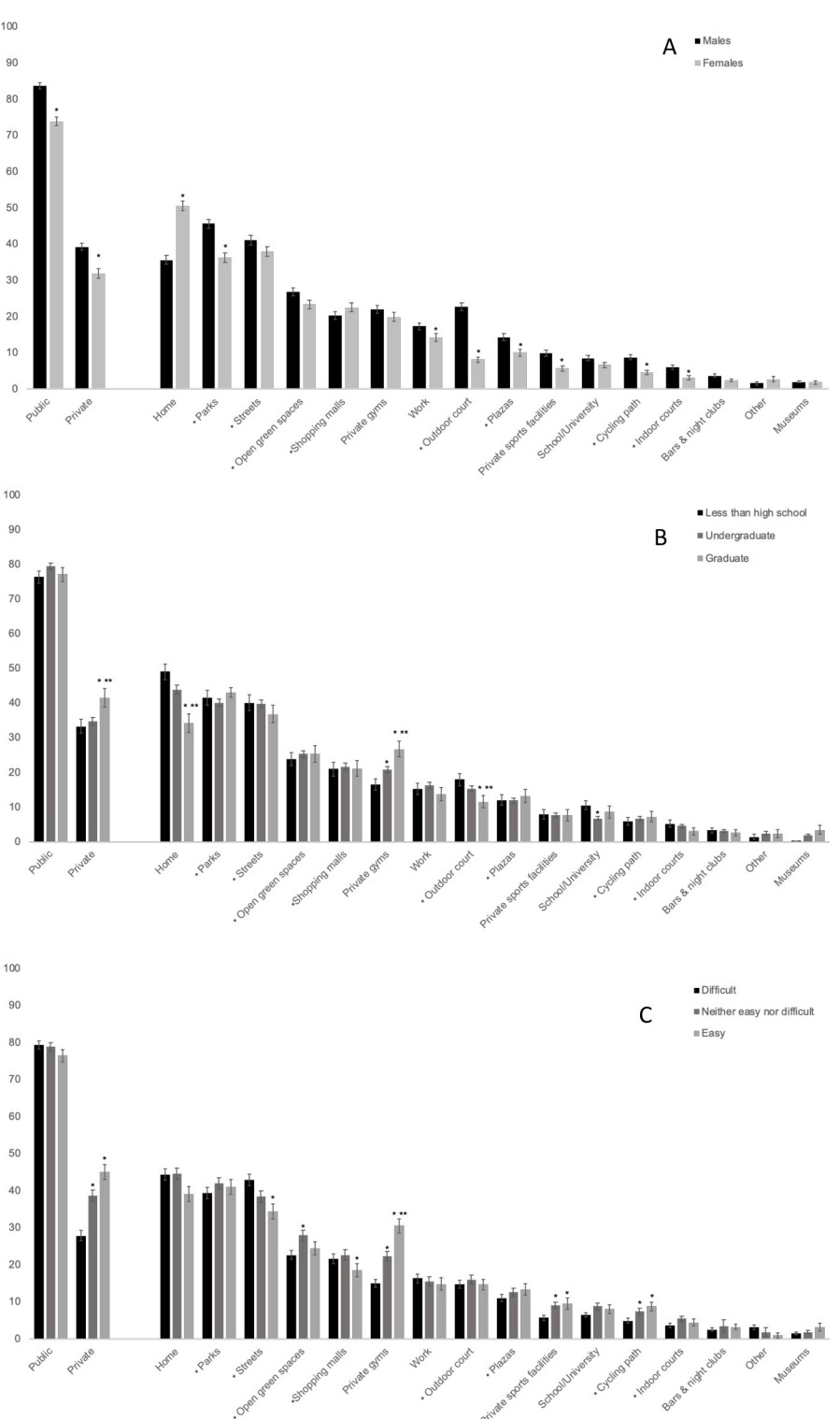

**Fig 2. Most frequently reported places for physical activity among adults from Mexico, across demographic characteristics (n = 3,686).** A. Gender: *Significantly different from males, B. Education level: *Significantly different from High school or less, **Significantly different from Undergraduate, C. Income adequacy: *Significantly different from Difficult, **Significantly different from Neither easy nor difficult • Indicates public space. Estimations (% and 95% CI) are weighted using survey weights.

**Table 3. Adjusted associations between places for physical activity and achieving physical activity recommendations among Mexican adults (n = 3,686).**

| Places for PA | Overall PA OR (95% CI) | Travel-related PA OR (95% CI)[e] | Leisure-time PA OR (95% CI)[e] | Work-related PA OR (95% CI)[e] |
|---|---|---|---|---|
| **Model 1 [c,d]** | | | | |
| Public-access place | **1.85 (1.49, 2.30)** | **2.41 (1.94, 2.99)** | 0.99 (0.80, 1.22) | 1.32 (1.04, 1.67) |
| Restricted-access place | **3.29 (2.67, 4.06)** | 0.98 (0.82, 1.18) | **4.36 (3.62, 5.25)** | 1.29 (1.06, 1.58) |
| **Model 2 [c]** | | | | |
| Home | **1.39 (1.15, 1.68)** | 1.17 (0.98, 1.38) | **1.45 (1.20, 1.75)** | 0.98 (0.81, 1.18) |
| Parks [a] | **2.31 (1.88, 2.83)** | **1.34 (1.12, 1.59)** | **1.91 (1.57, 2.32)** | 1.20 (0.99, 1.46) |
| Streets [a] | **1.49 (1.21, 1.82)** | **2.05 (1.71, 2,45)** | **0.73 (0.60, 0.89)** | 0.82 (0.67, 1.00) |
| Outdoors /open green spaces [a] | **1.34 (1.08, 1.66)** | 1.22 (1.00, 1.48) | 1.22 (0.98, 1.50) | 1.10 (0.89, 1.36) |
| Shopping malls [a] | **0.57 (0.45, 0.72)** | **0.71 (0.57, 0.89)** | **0.58 (0.45, 0.75)** | 1.17 (0.91, 1.51) |
| Private gyms [b] | **6.26 (4.52, 8.68)** | **0.77 (0.62, 0.96)** | **9.66 (7.34, 12.70)** | 1.07 (0.85, 1.42) |
| Work | **1.52 (1.15, 2.01)** | 1.25 (0.99, 1.59) | **0.45 (0.35, 0.59)** | **3.18 (2.49, 4.06)** |
| Outdoor court [a] | **2.97 (2.11, 4.20)** | 0.97 (0.77, 1.22) | **3.48 (2.61, 4.64)** | 1.12 (0.87, 1.43) |
| Plazas [a] | 0.99 (0.73, 1.36) | 1.20 (0.92, 1.57) | 0.72 (0.52, 0.98) | 1.25 (0.93, 1.67) |
| Private sports facilities [b] | **3.13 (1.85, 5.28)** | 1.26 (0.93, 1.70) | **5.03 (3.27, 7.74)** | 1.13 (0.81, 1.57) |
| School/University [b] | 1.37 (0.95, 1.97) | 1.20 (0.87, 1.66) | 0.91 (0.65, 1.28) | 1.43 (1.03, 1.99) |
| Cycling path [a] | **2.03 (1.26, 3.28)** | **1.91 (1.37, 2.68)** | 1.20 (0.76, 2.14) | 1.15 (0.83, 1.58) |
| Indoor courts [a] | 1.26 (0.69, 2.31) | 0.95 (0.64, 1.39) | 1.27 (0.76, 2.14) | 1.23 (0.81, 1.88) |
| Bars & night clubs [b] | 0.62 (0.40, 0.97) | 0.66 (0.43, 1.01) | 0.72 (0.46, 1.13) | 1.11 (0.70, 1.76) |
| Museums [b] | 0.73 (0.39, 1.38) | 1.35 (0.73, 2.51) | 0.62 (0.33, 1.18) | 0.97 (0.49, 1.90 |

PA = Physical activity

**Bolds** indicate significant (p<0.01) associations

a. Public-access place: no cost, membership or affiliation may be required for access and use.

b. Restricted-access place: cost, membership or affiliation required for access and use. Excludes home and work.

c. Models were adjusted for gender, age, education level, ethnicity, children living in the household, income adequacy, body mass index, and region of the country, and are weighted using post-stratification survey weights.

d. Models additionally adjusted for the use of home and work for physical activity

e. Domain-specific models were additionally adjusted for achieving 150 weekly minutes or more of the other two physical activity domains.

health-motivated PA behaviors (leisure-time activity) were more prevalent among those with higher income adequacy. One hypothesis that has been previously proposed, and that may help explain our findings, is the contrast of a choice- versus a necessity-driven PA model [18]. It has been suggested that it is only among populations where a substantial amount of people can afford owning a private vehicle, that travel-related PA is truly a choice. Conversely, this hypothesis proposes that among populations where the vast majority of people cannot afford owning a private motor vehicle, as in Mexico [9], transport-based PA is driven mainly by necessity. Although our sample seems to be more representative of highly educated and higher-income Mexicans where a choice-based model would be expected to operate, these data suggest that it is possible that both models operate in Mexico. The fact that our sample is composed of highly educated urban Mexicans likely reflects a different PA pattern that that of the overall population, possibly with higher leisure-time PA than those of the average population, and presumably, slightly lower levels of transport-based PA.

The choice- versus a necessity-driven PA model may also help explain the observed patterns of use of spaces. In our study, although public places were the most prevalent and relevant for travel-related PA, private places were most strongly associated with leisure-time PA. It is possible however that the relevance of private places for leisure-time PA among this sample (e.g.,

private gyms or sports clubs) is reflective of the fact that this is a highly educated sample, with higher access than the average population to these type of facilities. The use of streets as a place for PA was higher among those with lower income adequacy, whereas private gyms were more frequently used by those with a better income adequacy. Further, some public places (i.e. streets) were negatively associated with leisure-time activity (choice-based PA). This may be indicative of a substitution effect between PA domains: those with a higher income adequacy are able to choose to use a private place for PA, and thus, may generally not use places for transport-based PA or places for work-related PA, and vice versa. Future studies are needed to confirm our hypothesis. However, our results suggest that we may be upon a window of opportunity for PA promotion strategies seeking to engage all income levels in PA through public places. These strategies could foster a change in social norms related to leisure-time and travel-related PA. For example, according to our results, parks and outdoor courts are equally used by all education and income adequacy levels, and may contribute significantly to leisure-time activity (choice-based PA). Strategies for repurposing public spaces for PA, such as public programs providing free exercise classes in public-access places, could be effective for promoting PA in Mexico across socioeconomic strata, and may potentially contribute to changing the social norm related to places for PA [19].

Another effective strategy seeking to promote transport-based PA as a choice and change social norms related to travel-related PA may be already in place in Mexico. In line with previous work among users of the "*Muévete en Bici*" program (a Sunday Ciclovia program for leisure PA) in Mexico City [20], our results showed that cycling paths were more frequently used by participants with a better education or income adequacy (Fig 2) than their peers. One potential explanation for this is that in Mexico, efforts to increase cycling opportunities and infrastructure have been prioritized in high-income neighborhoods, and are generally not motivated by health promotion goals [21]. Rather, these efforts are driven by the need to reduce congestion and improve air quality in cities [21]. A study among Ecobici users (a public bicycle sharing program for transport) found that the main reason for program participation is travel efficiency [22]. The same study reported that participation in "*Muévete en bici*"(program for leisure PA) is mainly driven by health, wellness and enjoyment [22]. Future studies should explore the drivers of necessity vs. choice-driven bicycling in Mexico. The fact that these types of place-based activities are becoming popular among high-income groups may be a positive factor to capitalize on, as social norms on bicycling and the use of public places for PA may be starting to shift among Mexicans. Efforts to implement this type of infrastructure across neighborhoods of all income levels are warranted.

Consistent with previous reports [4], we found that the use of "informal" places, such as shopping malls and streets, was highly prevalent. The fact that these public places were reported as places for PA may help support previous hypotheses stating that social interaction may be an important driver for PA among Latin Americans [6]. Although we did not include a measure of social interaction in our study, in Mexico streets and shopping malls are commonly used as a place to hang out with friends. This may also help explain why the use of shopping malls as a place for PA was negatively associated with most PA outcomes. Although we did not measure the intensity of PA within specific places, the fact that in Mexico shopping malls are usual places for social interaction, walking at a leisurely pace (i.e. low intensity) is a common PA occurring in malls. Evidence continues to build up supporting the role of light-intensity PA for health, particularly for highly inactive groups and for older adults [23]. Some evidence from high-income countries also supports the relevance of shopping malls for place-based PA interventions, such as walking programs targeted at among middle-aged and older adults [24]. Given that the presence of shopping malls is growing at an unprecedented rate in Mexico and Latin America, they may represent an opportunity to promote walking at the population level.

Two of our findings also highlight potential opportunities for PA promotion in Mexico. First, the workplace was only associated with work-related PA. Although this may seem obvious, it highlights the fact that in Mexico no efforts are being made to promote leisure-time PA within the workplace. Second, women's use of most of the studied places for PA was lower compared to men. It has been proposed that societal norms may play an important role in the underlying causes of this gender-based disparity [25], underscoring the relevance of gender-based approaches for PA promotion.

To our knowledge, this is the first study examining the contribution of places and PA domains to overall PA among a sample of adults from all over Mexico. However, our study had limitations. Although being of national coverage and utilizing complex sampling weights, our sample is not representative at the national level. The use of an online survey might have biased the findings in Mexico as 40% of the Mexican population does not have internet access and internet is often restricted to certain residential and urban areas [26]. Also, the Mexico sample had higher levels of education and lower levels of BMI compared to national estimates [9,10]. Therefore, the external validity of our findings is limited to those with similar characteristics to our sample. However, we believe that our results provide important insights for understanding the drivers of PA behaviors among highly educated (and probably high income) groups in Mexico, which until now have been understudied. In this study, we did not measure household car ownership, which would have allowed us to better examine the role of necessity-versus choice-based PA among this sample.

## Conclusion

In our sample, travel-related and leisure-based activity were major contributors for overall PA. The widespread use of public-access places for PA (parks, streets, open green spaces and outdoor courts) across socioeconomic strata, and their significant contribution towards meeting PA recommendations among participants, suggests these places may be powerful settings for implementing PA promotion strategies in Mexico. On the other hand, policies geared at increasing the equitable access for all to what currently are "private" places (i.e. private gyms and sport facilities) may also represent an effective strategy to increase PA among Mexican adults, as these places were strongly associated with PA in our sample [6].

## Supporting information

**S1 Table. Most frequently reported places for physical activity (PA) among males and females from Mexico (n = 3,686).**
(DOCX)

**S2 Table. Most frequently reported places for physical activity (PA) across education levels among adults from Mexico (n = 3,686).**
(DOCX)

**S3 Table. Most frequent places for physical activity across categories of income adequacy among Mexican adults (n = 3,686).**
(DOCX)

## Author Contributions

**Conceptualization:** Alejandra Jáuregui, Deborah Salvo, Catalina Medina.

**Data curation:** Alejandra Jáuregui.

**Formal analysis:** Alejandra Jáuregui.

**Funding acquisition:** David Hammond.

**Investigation:** David Hammond.

**Methodology:** David Hammond.

**Project administration:** David Hammond.

**Resources:** David Hammond.

**Supervision:** Simón Barquera.

**Writing – original draft:** Alejandra Jáuregui.

**Writing – review & editing:** Deborah Salvo, Catalina Medina, Simón Barquera, David Hammond.

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
