## [Decision Letter · Decision Letter 0]

6 Jan 2020

PONE-D-19-30067

Understanding the contribution of public- and restricted-access places to overall and domain-specific physical activity among Mexican adults: A cross-sectional study.

PLOS ONE

Dear Dr. Jáuregui,

Thank you for submitting your manuscript to PLOS ONE. After careful consideration, we feel that it has merit but does not fully meet PLOS ONE’s publication criteria as it currently stands. Therefore, we invite you to submit a revised version of the manuscript that addresses the points raised during the review process.

We would appreciate receiving your revised manuscript by Feb 20 2020 11:59PM. To enhance the reproducibility of your results, we recommend that if applicable you deposit your laboratory protocols in protocols.io, where a protocol can be assigned its own identifier (DOI) such that it can be cited independently in the future. For instructions see: http://journals.plos.org/plosone/s/submission-guidelines#loc-laboratory-protocols

We look forward to receiving your revised manuscript.

Kind regards,

Anne Vuillemin

Academic Editor

PLOS ONE

Journal Requirements:

Reviewers' comments:

Reviewer's Responses to Questions

**Comments to the Author**

1. Is the manuscript technically sound, and do the data support the conclusions?

Reviewer #1: Yes

2. Has the statistical analysis been performed appropriately and rigorously? 

Reviewer #1: Yes

3. Have the authors made all data underlying the findings in their manuscript fully available?

Reviewer #1: Yes

4. Is the manuscript presented in an intelligible fashion and written in standard English?

Reviewer #1: Yes

5. Review Comments to the Author

Reviewer #1: The manuscript is well-written, very concise and informative for providing evidence to inform the design of public strategies. There are some minor comments and observations for the paper. The discussion is very well-developed and it is easy to follow. This section adds relevant and contextual information about the implications of these findings.

Introduction.

L77. In the paper is mentioned that a limitation of current studies is that they only included one city from each studied country… could you provide in methods (maybe in the discussion too) briefly more details about the included cities in the study. It could be relevant for international readers.

Methods.

L110. Please check the reference, as the criterion validity was not tested in Brazil. Also, in general, the paper describes that reliability was moderate and validity poor to fair in other countries.

L114. Add at the end of the sentence in the parenthesis… “per week of MVPA).”

Statistical analysis. In the abstract, you mentioned the analytical method used in this paper, but it was not mentioned in this section.

Results

How did you evaluate implausible or inconsistent results? (you may add details in methods).

L181. For international comparability, you may use the term university instead of college as in some countries that refer to high school.

L213. minor detail. use PA instead of physical activity.

L225. minor detail. add % after 29 to be consistent.

Discussion

In the discussion, please describe the implications of having a more educated sample in this study compared with the Mexican population. You may add some examples or hypothesise something. I know you highlight this in the limitations (L347).

Tables and figures

I may include the supplementary tables in the paper as you described them in the results. It would be very useful to see the sex differences in the full version of the manuscript, for example.

6. PLOS authors have the option to publish the peer review history of their article (what does this mean?). If published, this will include your full peer review and any attached files.

Reviewer #1: No

---

## [Author Response · Author response to Decision Letter 0]

12 Jan 2020

Reviewer #1: The manuscript is well-written, very concise and informative for providing evidence to inform the design of public strategies. There are some minor comments and observations for the paper. The discussion is very well-developed and it is easy to follow. This section adds relevant and contextual information about the implications of these findings.

Introduction.

L77. In the paper is mentioned that a limitation of current studies is that they only included one city from each studied country… could you provide in methods (maybe in the discussion too) briefly more details about the included cities in the study. It could be relevant for international readers.

Response: Thank you for your suggestion. We have now detailed the cities included in each of the three countries in the introduction and in the discussion sections (L77-78 and L272-274). 

Methods.

L110. Please check the reference, as the criterion validity was not tested in Brazil. Also, in general, the paper describes that reliability was moderate and validity poor to fair in other countries.

Response: We thank the reviewer for pointing out this mistake. We have now included addittional references reporting on the criterion validity of GPAQ among Latinamerican populations and corrected the sentence reporting the validity of the GPAQ to reflect that this instrument has poor to fair validity (L112-113) . 

L114. Add at the end of the sentence in the parenthesis… “per week of MVPA).”

Response: Done

Statistical analysis. In the abstract, you mentioned the analytical method used in this paper, but it was not mentioned in this section.

Response: We thank the reviewer for identifying this omission. We have now included in this section the type of models used (i.e. multivariate logistic regression models) (L163). 

Results

How did you evaluate implausible or inconsistent results? (you may add details in methods).

Response: Implausible and inconsistent PA data was evaluated according to the GPAQ protocol. We have now included additional information and examples of how this was achieved in the “Physical activity measures” section (L114-116). 

L181. For international comparability, you may use the term university instead of college as in some countries that refer to high school.

Response: Done

L213. minor detail. use PA instead of physical activity.

Response: Corrected

L225. minor detail. add % after 29 to be consistent.

Response: Corrected

Discussion

In the discussion, please describe the implications of having a more educated sample in this study compared with the Mexican population. You may add some examples or hypothesise something. I know you highlight this in the limitations (L347).

Response: Wee thank the reviewer for the suggestion. We have now included additional information relating the implications of having a more educated sample in the study compared with the Mexican popualtion (L287-290). 

Tables and figures

I may include the supplementary tables in the paper as you described them in the results. It would be very useful to see the sex differences in the full version of the manuscript, for example.

Response: We thank the reviewer for the suggestion. We have now included an additional figure showing differences by demographic characteristics (i.e. sex, education and income adequacy) (see Figure 2, L218-220). We decided to keep supplementary tables (i.e. S1, S2 and S3 Table) because they provide additional information which may be useful for readers (L210-211).

---

## [Editor Report · Decision Letter 1]

17 Jan 2020

Understanding the contribution of public- and restricted-access places to overall and domain-specific physical activity among Mexican adults: A cross-sectional study.

PONE-D-19-30067R1

Dear Dr. Jáuregui,

We are pleased to inform you that your manuscript has been judged scientifically suitable for publication and will be formally accepted for publication once it complies with all outstanding technical requirements.

With kind regards,

Anne Vuillemin

Academic Editor

PLOS ONE
---

## [Editor Report · Acceptance letter]

27 Jan 2020

PONE-D-19-30067R1 

Understanding the contribution of public- and restricted-access places to overall and domain-specific physical activity among Mexican adults: A cross-sectional study. 

Dear Dr. Jáuregui:

I am pleased to inform you that your manuscript has been deemed suitable for publication in PLOS ONE. Congratulations! Your manuscript is now with our production department. 

With kind regards,

on behalf of

Dr. Anne Vuillemin 

Academic Editor

PLOS ONE